# Predicting Parent Trust Based on Professionals' Communication Skills

James Edward Hamm * and Angela DeSilva Mousseau

Department of Education and Counseling, Benoit Education Center, Rivier University, 420 South Main Street, Nashua, NH 03060, USA
* Correspondence: jhamm@rivier.edu; Tel.: +1-603-897-8282

**Abstract:** Trust is critical to the establishment and maintenance of working relationships between the parents of children with disabilities and their child's professional. Knowledge of the specific communication skills needed to secure trust is unclear. The current study investigated the relationship between parent evaluation of professionals' communication skills and parent trust of professionals. A total of 165 parents responded to an online survey during the COVID-19 pandemic. The results indicated that professionals' communication skills had a significant and moderately positive relationship with the parent trust of professionals. Hierarchical multiple regression analyses indicated that parents' ratings of professionals' communication competence, and professionals' use of in-person communication were the only predictors of parent trust of special education professionals, even when other factors were considered. This study's findings draw attention to the importance communication skills may have in establishing and maintaining trusting relationships with parents.

**Keywords:** parents of children with disabilities; communication skills; communication methods; trust; special education professionals

## 1. Introduction

The Individuals with Disabilities Education Act (IDEA; 2004) is a law that was written to improve the outcomes of students with disabilities living in the United States. Students who might otherwise be segregated from non-disabled peers are granted access to a free and appropriate public education (FAPE). An important accountability component of this law mandates parent involvement in the planning and programming of a student's individualized education program (IEP). IDEA, furthermore, assures that districts will inform parents regularly about the progress their child has made toward annual goals [1].

In as much as districts attempt to comply with IDEA in their communication with parents, disagreements about the design and delivery of instruction occur [2]. If unresolved, disagreements can lead to third-party involvement outside of team processes. Due process complaints and litigation, team mediation, and meeting facilitation exist to remedy conflicts under IDEA. Due process complaints and litigation are adversarial, time consuming [2–4], and strain the financial and emotional resources of parents and schools [3–7]. Mediation and meeting facilitation can help repair strained parent–professional relationships [3,8,9], but require the skills and resources of participants outside the conflict [2,10]. To improve the efficiency with which parent–teacher conflicts are resolved, it is worth considering whether there are skills that special education professionals can develop and enact to reduce the negative outcomes associated with parent–teacher conflicts.

### 1.1. Parent–Teacher Conflict and Trust

Understanding parent–teacher conflict requires an understanding of its' contributing factors. Lake and Billingsley [11] studied parents and educators who were involved in the special education appeals process. They found that in addition to fiscal constraints, service delivery, and parent knowledge, the presence of individual communication patterns and trust influenced whether a conflict escalated or de-escalated. Parent perceptions of

a professional's communication effectiveness, communication frequency, listening skills and honesty contributed to or reduced a conflict. When parents trusted their child's professionals, they were more tolerant when professionals shared information that might be deemed problematic. When parents did not trust their child's professional, they made fewer attempts to communicate, expected fewer positive outcomes and lacked confidence in a professional's "good faith efforts". Studies have also suggested that trust and communication are factors that parents consider when deciding the actions they take in their interactions with their child's professional, such as deciding whether to use an advocate or not [12–14].

Based on Lake and Billingsley's research [11], trust is a factor that must be considered if special education professionals are to succeed at establishing and maintaining working relationships with parents. Efforts to define trust, however, have been fraught with a lack of consensus among various social science disciplines [15–17]. Within educational contexts, attempts to define trust have resulted in multiple definitions as well. Attempts to study it empirically have been undertaken with multifaceted measures that involve multiple subcategories [18–24], and singular definitions with a one-dimensional measure [25,26]. Tschannen-Moran and Hoy [18], for example, developed a multifaceted measure of trust that evaluates an individual's willingness to be vulnerable to risk based on perceptions of another's benevolence, reliability, competence, and openness. Tschannen and Hoy's measures have been used in the study of collective properties of trust relations within school communities [19,27–29]. Adams and Christianson [26], on the other hand, developed the Family–School Relationship Survey (FSRS) to evaluate the parent trust perceptions of teachers, and conceptualized this form of trust as " . . . confidence that another person will act in a way to benefit or sustain the relationship, or the implicit or explicit goals of the relationship, to achieve positive outcomes for students." [26] p. 480. In both instances, when parent trust is present it can portend positive outcomes for teachers and schools. Parent trust has been associated with academic achievement [21], enabling school structures where school members are inclined to engage in problem-solving [28], and parent involvement in schools [26,30]. None of the above studies, however, collected information on students with disabilities or their parents.

When the parent trust of students with disabilities has been evaluated, it has been undertaken with qualitative designs [31–33] or designs that have examined it as an inseparable component of parent-partnerships [34]. What is more, no studies thus far have examined the parent trust of students with disabilities as an outcome variable, or a property that is to be desired on behalf of the efforts of special education professionals.

*1.2. Interacting with Parents Effectively and Appropriately*

Determining effective and appropriate methods for interacting with parents of students with disabilities has been an important undertaking in the special education literature. The research suggests that parents value a professional's expression of clinical knowledge [11,35–37]. They also appreciate the professional's understanding of a child's and family needs [11,38], and their receptivity to parent input [39–41]. Parents also prefer communication that is frequent, honest, open, informal, and genuine [13,17,42]. They disapprove of interactions in which they perceive that paperwork and educational jargon are the primary topics of conversation rather than their child's educational needs [42–44].

While the current literature provides general descriptions of how parents expect their professionals to communicate, it lacks the specificity professionals need to interact effectively or appropriately with parents. It is also unclear whether enacting specific behaviors can predict trust, which has been identified as an important outcome in the literature. The aim of our quantitative survey study was, therefore, to investigate the relationship between the parent impressions of special education professionals' communication skills and the parent trust of students with disabilities. To understand this relationship more fully, we sought to answer the research question of what factors potentially impact trust

between parents and special education professionals, and proposed the following null and research hypotheses:

**Hypotheses 1.** *There is not a relationship between a professional's communication skills and the trust of parents of students with disabilities.*

**Hypotheses 2.** *There is a relationship between a professional's communication skills and the trust of parents of students with disabilities.*

We set out to use a hierarchical multiple regression design to understand this relationship. We selected Spitzberg's [45] Conversational Skills Rating Scale (CSRS) to serve as our independent variable as a measure of parents' perceptions of special education professionals' communication skills, and a modified version of Adams and Christenson's [26] Family–School Relationship Survey (FSRS) as our dependent variable trust. We also attempted to collect information regarding demographic variables that may influence parent levels of trust.

## 2. Materials and Methods

Following internal review board (IRB) approval from the authors' institution, a sample of 165 parents from across the United States was obtained. Approval from the OSEP-funded support group over email communication was sought from the group's director. An advertisement and link were created for the director to approve and administer to group members over email. An advertisement using the same wording was created and administered over Facebook that targeted parents of children with disabilities. The advertisement indicated that the study was about the relationship between the parents of students with disabilities and professionals who are responsible for contacting parents and ensuring that the IEP services are provided. A description of the inclusion and exclusion criteria, and the approximate time it would take to complete the survey (i.e., approximately 15 min) were included in the advertisement. In order to be selected for the study, the parents had to live in the United States, have a child identified with a disability, and have a child who received services through an individualized education program (IEP). The parents were also required to have had at least one interaction with a professional to evaluate the competence of the professional's communication. Participants who entered the study were asked to read and complete the consent form. This form included a brief introduction, a description of the research, the benefits of the study, its risks, a confidentiality statement, an offer of a potential to earn a USD 50 Amazon gift card from a raffle, and buttons to press to 'accept' or 'decline' to enter the study. In terms of the demographic variables, we asked the parents about their age, parent role, marriage status, employment status, race, whether their child received free and reduced lunch as a proxy for socioeconomic status, the age of their child, age at which their child began receiving services, the length of time they had received services from their assigned professional, their child's identified disability, school description, and child's learning format. This study was conducted during the latter half of a global pandemic; specifically, from November of 2021 through January of 2022.

### 2.1. Participants
#### 2.1.1. Parents

Parents who responded to the questionnaires ranged in age from 23 to 63 (M = 41). Most were mothers (n = 134) (61%), but fathers (n = 25; 15%), grandmothers (n = 3; 2%), guardians (n = 2) (1%), and a stepmother (0.6%) were also represented. Most parents indicated that they were married or in domestic partnerships (n = 140) (84.8%). Fourteen (8.4%) parents reported that they were single or never married, and eleven reported that they were divorced (6.7%). Eighty-eight parents indicated that they were employed full-time (53%), whereas others reported that they were employed part-time (n = 42; 25.45%), or unemployed (n = 35; 21.21%). Most parents indicated they were white (n = 112) (67.87%). The study included a smaller number of participants who indicated they were Hispanic/Latino/a (n = 10; 6%), American Indian or Alaska Native (n = 9; 5.45%), Asian (n = 3; 1.81%), Black

or African American (n = 19; 11.52%), Native Hawaiian or other Pacific Islander (n = 2; 1.21%), and individuals identified as being two or more races (n = 10; 6%). Eighty-three parents (49%) indicated that they worked in education, while eighty-one parents (49%) reported that they had not worked in education. Parent education ranged from 11 years to 23 years, with a mean of 16.48 years. This value is approximately the equivalent of a four-year college degree. A free and reduced lunch price was used as a proxy for lower socioeconomic status (SES). Sixty (36.36%) parents indicated that their child received a free or reduced lunch price; 105 parents (63.63%) indicated their child did not receive a free or reduced lunch price. Table 1 summarizes the demographic information we obtained from parents regarding parent role, marriage status, employment status, race, prior work experiences in education, and socioeconomic status.

**Table 1.** Frequencies and Percentages of Parent Characteristics.

| Demographic Category | Characteristic | Frequency | Percentage |
|---|---|---|---|
| Parent Role | Mother | 134 | 81.2 |
| | Father | 25 | 15.2 |
| | Stepmother | 1 | 0.6 |
| | Grandmother | 3 | 1.8 |
| | Guardian | 2 | 1.2 |
| Marriage Status | Single, never married | 14 | 8.5 |
| | Married | 140 | 84.8 |
| | Divorced | 11 | 6.7 |
| Employment Status | Not employed | 35 | 21.2 |
| | Part-time employed | 42 | 25.5 |
| | Employed full-time | 88 | 53.3 |
| Race | Hispanic/Latino/a | 10 | 6.1 |
| | American Indian/Alaska Native | 9 | 5.5 |
| | Asian | 3 | 1.8 |
| | Black or African American | 19 | 11.5 |
| | Native Hawaiian or Other Pacific Islander | 2 | 1.2 |
| | White | 112 | 67.9 |
| | Two or more races | 10 | 6.1 |
| Prior work experiences in education | Yes | 83 | 49 |
| | No | 81 | 49 |
| Socioeconomic Status | Not Low | 105 | 63.6 |
| | Low | 60 | 36.4 |

Parents from 30 states, and from four separate regions entered the study. Ninety (54.54%) indicated they were from the northeast, thirty (18.79%) from southern states, fifteen (9.09%) from a midwestern state and twenty-nine (15.58%) from a western state. Table 2 represents the U.S. regions in which parents live. The top five states represented in the sample were Massachusetts (n = 34; 20.6%), New Hampshire (n = 29; 17.58%), California (n = 19; 11.52%), Texas (n =10; 6.06%), and New York (n = 9; 5.45%).

**Table 2.** Parent participants by U.S. region.

| Category | Characteristic | Frequency | Percentage |
|---|---|---|---|
| U. S. region | Northeast | 90 | 54.55 |
| | South | 31 | 18.79 |
| | Midwest | 15 | 9.09 |
| | West | 29 | 17.58 |

2.1.2. Special Education Professionals

Parents were asked to indicate the job title of their child's assigned professional who was responsible for contacting them about their child's IEP. Special education teachers

most frequently performed this duty (n = 98; 59.4%), followed by case managers/liaisons (n = 20; 12.1%), speech/language pathologists (n = 12; 7.3%), special education coordinators (n = 9; 5.5%) school psychologists (n = 7; 4.2%), occupational therapists (n = 6; 3.6%), and other educational professionals (n = 10; 6.1%). Of these professionals, the parents reported that their child's professional worked simultaneously as their child's special education teacher (n = 123) in 74.5% of cases. Forty-one parents (24.8%) indicated that the professional responsible for contacting them was not a special education teacher. One parent did not provide responses to these items (0.6%). Ninety-seven parents (58.8%) indicated that the special education professional had worked with their child during the pandemic, and sixty-eight (41.2%) reported that the special education professional did not work with their child during the pandemic.

### 2.1.3. Child Characteristics

The children ranged in age from three to twenty-one years (M = 10; SD = 4.001). The age at which schools identified children with a disability ranged from infancy to fourteen years (M = 4; SD = 3.093). The length of time that the currently assigned professionals provided services to the children ranged from less than one month to eleven years and six months (M = 1 year and 9 months; SD = 3 years and 8 months). The children received between two months to twenty years and ten months of special education services (M = 3 years and 9 months). All disability categories that were available for parents to select were identified in the sample. We selected the disability categories based on the state level disability categories in which the author's work. These included children identified with autism (n = 53), a communication impairment (n = 49), a developmental delay (n = 46), an emotional disturbance (n = 22), a specific learning disability (n = 53), a health disability (n = 63), an intellectual disability (n = 24), a neurological disability (n = 22), or a sensory impairment (n = 23). Most children attended public schools (n = 129; 78.2%). Fewer numbers of children attended private (n = 28; 17.0%), and charter schools (n = 8; 4.8%). Most children attended school in-person (n = 141; 85.5%). Fewer numbers attended school remotely (n = 12; 7.3%), or in a hybrid format (n = 12; 7.3%). Table 3 provides information regarding the school descriptions and learning formats of children.

**Table 3.** School description and learning formats of children.

| Demographic Category | Characteristic | Frequency | Percentage |
|---|---|---|---|
| School Description | Public | 129 | 78.2 |
| | Private | 28 | 17 |
| | Charter | 8 | 4.8 |
| Learning Format | In-person | 141 | 85.5 |
| | Remote | 12 | 7.3 |
| | Hybrid | 12 | 7.3 |

### 2.1.4. Data Collection

We used a demographic questionnaire to gather information about the parents, their children and their child's professional. Regarding the children, we asked for the length of time the children had received services overall, the length of time that they received services from their current professional, their disability category, their school placement, and their learning format (e.g., remote, in-person, or hybrid). For the parents, we asked for details regarding their age, role, marriage status, state of residence, and education attainment. With respect to the professionals, we asked about their job title, and the frequency of communication with a professional. We also asked two questions that pertained to COVID-19. The first question asked, "Did the special education professional who is assigned to you and your child work with you and your child during the 2020–2021 school year (during the COVID-19 pandemic)?". The parents could answer either "yes" or "no" to this question. The second question asked parents about how COVID-19 impacted their relationship with their child's professional. Specifically, the parents were asked, "How has the COVID-

19 pandemic impacted your relationship with the special education professional who is responsible for contacting you?". The parents could respond to this question with the following answers: "positively", "negatively", "not at all", or "unable to assess". We also asked the parents whether their child's professional communicated with them in-person, or through other modes of communication such as phone, video conferencing, email, text messaging, or notes. The parents were also asked to identify whether they used services outside the school such as lawyers or advocates, and whether they used legal measures to obtain services for their children, such as mediation, meeting facilitation, or due process complaints or litigation. The parents were then asked to think about a conversation they had with their professional and respond to two questionnaires that rated the impressions of their professional's communication skills and the trust of their professional. We performed our analyses using IBM SPSS Statistics version 28.0.

We used the Conversational Skills Rating Scale (CSRS) as our independent variable to measure the parent perceptions of their professional's communication skills. The CSRS is a 30-item scale developed by Spitzberg [45] that assesses the competence of an individual's communication skills. Skills are " ... viewed as reproducible, goal-directed, functional actions and action sequences [that, by definition, are] observable, relatively discrete, and operational." [45] p. 11. The CSRS is divided into 25 items. These items refer to specific behaviors, such as the use of eye-contact, asking questions, posture, and the use of gestures. The CSRS was designed to measure four skill domains including attentiveness, composure, expressiveness, and coordination. Spitzberg [45] reports that the CSRS has demonstrated internal reliability (i.e., a coefficient alpha above 0.85) and reliability of factor subscales (>0.80). We asked the participants to imagine a conversation they had with their professional when responding to the CSRS. Sutherland and Yoshida [46] used the CSRS in a similar way when they examined the relationship between teachers' perceptions of leaders' communication skills and the teacher trust of leaders. They reported that the CSRS had strong reliability at the subcategory level (i.e., attentiveness-coordination $\alpha = 0.95$, composure $\alpha = 0.95$, and expressiveness $\alpha = 0.92$). Our use of Pearson product-moment correlations revealed that the attentiveness, composure, expressiveness, and coordination subcategories all had moderately strong correlations with one another. This outcome led us to use an overall score of communication skills as our independent variable. The CSRS items were formatted as a 5-point Likert scale. The parents read items and rated their professionals as having 1 = "Inadequate", 2 = "Fair", 3 = "Adequate", 4 = "Good", and 5 = "Excellent" communication skills. The total scores of this measure ranged from 25 to 125. Our analysis showed that the CSRS had an excellent reliability as a general measure of communication skills (CSRS, $\alpha = 0.97$).

To measure the parent trust of special education professionals as our dependent variable, we used an adapted version of Adams and Christenson's [26] Family–School Relationship Survey (FSRS). The FSRS was initially designed to evaluate the parent trust of teachers. It was found to be a reliable measure of parent trust in Adams and Christenson's study [26] ($\alpha = 0.96$), and another study by Santiago et. al. [30] ($\alpha = 0.92$). We altered the items to refer to the trust of special education professionals. For example, the statement, "I am confident that teachers are receptive to my input and suggestions", was modified to, "I am confident that my special education professional is receptive to my input and suggestions". The definition of trust used in the study was " ... confidence that another person will act in a way to benefit or sustain the relationship, or the implicit or explicit goals of the relationship, to achieve positive outcomes for students." [27] p. 480. The FSRS had an excellent reliability in our study ($\alpha = 0.98$).

### 2.2. Data Analysis

We anticipated using a hierarchical multiple regression to analyze the relationship between the independent variables that included the demographic characteristics, communication patterns, and parent perceptions of professionals' communication skills and their potential impact on the dependent variable trust. In our first level of analysis, it was

important to establish which independent variables in our study were related to or revealed differences in the trust of the parents. Pearson product-moment correlation statistics were run to determine the extent to which the independent variables were related to trust, and independent samples *t*-tests and one-way ANOVAs were used to determine significant differences of trust between the various parent groups. If we found variables that related to or revealed significant differences, we would add them to our regression model. We expected that a professional's interpersonal communication skills would predict parent trust, above and beyond the demographic characteristics. If these expectations were met, it was anticipated that the research hypothesis would be accepted, and the null hypothesis would be rejected.

## 3. Results

### *3.1. The Impact of Demographic Characteristics on Trust*

3.1.1. Demographic Characteristics of CSRS and FSRS

Scores ranged from 27 to 112 (M = 85.45, and SD = 19.62) on the CSRS, and scores ranged from 19 to 76 (M = 55.51, and SD = 14.57) on the FSRS.

3.1.2. Family Demographic Characteristics, School Type, Learning Formats, and Professional's Job Description and Parent Trust

The parent trust of students' special education professionals did not relate to the type of school the students attended, the learning format of the school or the job title of the student's special education professional.

3.1.3. The Impact of a Child's Disability Category on Parent Trust

A series of independent *t*-tests were run to determine whether differences in trust levels existed between the parents based on disability categories. These categories included specific learning disabilities (SLD) (e.g., reading), autism, communication impairments, health disabilities (e.g., ADHD), emotional impairments, developmental delays, intellectual and neurological disabilities, and sensory impairments (e.g., vision). The only significant differences we found based on group identification included parents who did and did not indicate that their child had a specific learning disability. Specifically, the parents who indicated that their child had an SLD trusted their professional (M = 52.26) significantly less than the parents of children not identified with SLDs (M = 57.09) (t = 2.06, df = 163, and *p* = 0.045, two tailed). The effect size of this difference was small (d = 0.34). Again, significant differences were not detected based on the other disability categories.

### *3.2. COVID-19's Impact on Parent Trust*

3.2.1. Parent Trust of Professionals Who Worked with Them during the Pandemic

Parents who indicated that they worked with their child's professional during the pandemic expressed higher levels of trust in their child's professional (M = 57.55) than those parents who did not work with their child's professional during the pandemic (M = 52.60). An independent *t*-test indicated a significant difference between these groups (t = 2.17, df = 163, and *p* = 0.032, two tailed). The effect size of this difference was small (d = 0.34).

3.2.2. Parent Trust Based on COVID-19's Impact on Communication

A one-way between subjects ANOVA indicated significant differences in trust between the groups of parents who were asked to indicate the pandemic's impact on their communication (F(3,161) = 4.403, and *p* = 0.005). A Tukey HSD test for multiple comparisons indicated that the mean value of trust was significantly different between parents who reported that they were "unable to assess" COVID-19's impact, and those parents who reported that the COVID-19 pandemic impacted their communication "positively" (*p* = 0.004, and a 95% confidence interval = 2.90, 21.34). The parents who were able to assess the impact of COVID-19 on their communication also differed in their reported levels of

trust from those parents who reported that COVID-19 did not impact their communication with their child's professional ($p$ = 0.044, and a 95% confidence interval = 0.17, 16.69).

We used dummy variables to run independent *t*-tests to determine whether parents of the positive impact group differed in their levels of trust from parents who did not report a positive impact. Those who selected positive were coded with a 1, and those who did not indicate that COVID-19 had a positive impact were coded with a 0. The findings indicated that the parents who reported that COVID-19 had a positive impact on their communication trusted their child's professional to a significantly greater extent than those parents who did not report that COVID-19 had a positive impact (positive: M = 60.74; not positive M = 54.10) (trust: t = 2.43, df = 163, and $p$ = 0.016). This difference had a small effect size (d = 0.46).

We also used dummy-coded variables to run independent *t*-tests that determined whether parents of the "able to assess" impact group differed in their levels of trust from parents who were not able to assess the impact of COVID-19. Those who were able to assess the impact of COVID-19 were coded with a 1, and those who were unable to assess the impact of COVID-19 were coded with a 0. The results indicated that the parents who were able to assess the impact of COVID-19 trusted their professionals significantly more than those parents who were unable to assess the impact of COVID-19 (able to assess COVID-19 impact M = 56.98; not able to assess COVID-19 impact M = 48.62) (t = 2.87, df = 163, and $p$ = 0.005). The effect size of this difference was moderate (d = 0.59).

### 3.3. Methods of Communication and Parent Trust

### 3.3.1. The Relationship between Communication Frequency and Parent Trust

A Pearson product-moment correlation showed that the frequency with which parents communicated with their child's assigned professional was not related to the parent trust of their professional (r = 0.15, N = 164, and $p$ = 0.062, two tailed).

### 3.3.2. In-Person Communication

Parents indicating that they spoke with their child's professional in-person expressed higher levels of trust in their professional (M = 59.80) than parents who did not speak in-person with their child's professional (M = 53.06). An independent *t*-test showed that the difference between these groups was significant, (t = 3.14, df = 149.54, and $p$ = 0.002, two tailed). The effect size of this difference was small (d = 0.47).

### 3.3.3. Distance Communication

Independent *t*-tests were run to determine whether differences in trust could be detected based on various forms of distance communication. These categories included professionals who had phone conversations, held video conferences, used text messaging, or used a communication notebook to contact parents. Phone communication emerged as the only factor that significantly differentiated the trust between parents. Specifically, parents trusted their child's professionals significantly more who contacted them over the phone (M = 57.94) than those parents whose professionals did not contact them over the phone (M = 52.7) (t = 2.28, df = 162, and $p$ = 0.004, two tailed). The effect size of this difference was small (d = 0.36). Significant differences were not found based on other forms of distance communication.

### 3.4. Independent Services, Legal Measures, and Parent Trust

### 3.4.1. Parent Use of an Independent Service

An independent *t*-test indicated that parents who used a service independent of their child's school services (e.g., an advocate, independent evaluator, or lawyer) expressed significantly lower levels of trust (M = 53.23) than those parents who did not use an outside service (M = 58.70) (t = 2.42, df = 163, and $p$ = 0.017, two tailed). The effect size of this difference was small (d = 0.38).

### 3.4.2. Legal Measures to Obtain Services

An independent *t*-test indicated that the parents who used legal measures (e.g., mediation, meeting facilitation, due process complaints and litigation) to obtain services for their child expressed lower levels of trust (M = 52.43) than those parents who indicated that they did not use legal services (M = 58.13) (t = 2.55, df = 163, and *p* = 0.012, two tailed). The effect size of this difference was small (d = 0.40).

### 3.5. Communication Skills and Trust

A Pearson product-moment correlation was run to determine the strength of relationship between the parent ratings of their professional's communication skills and the parent trust of the professional. A significant and moderately strong relationship was found between these variables (r = 0.66, N = 165, and *p* < 0.001, two tailed).

### 3.6. Primary Analysis: Regression

A hierarchical multiple regression was run to determine the collective and separate effects that the significant independent variables had on the parent trust of professionals. A preliminary analysis indicated no violations of linearity, multicollinearity, independence of observations, homoscedasticity, or normality. Case-wise diagnostics uncovered seven outliers that were three standard deviations above the mean trust score. These observations were removed from further analyses. All indicated that they had used an independent evaluator to secure services for their child, and all seven indicated that they worked with their professionals during the pandemic, but other common characteristics could not be determined from the data we obtained. The independent variables were entered in four separate blocks. The variables included in Block 1 were COVID-19 variables that differentiated the trust levels of parents. Block 2 added the demographic variables found to significantly differentiate the trust levels of parents onto the COVID-19 variables in Block 1. Block 3 added the professional communication methods that were found to significantly differentiate the trust levels of parents onto the significant COVID-19 variables, and the significant parent demographic variables from Blocks 1 and 2. Block 4 included the communication skills of professionals onto the significant variables from blocks 1, 2, and 3. Table 4 provides a reference of variables as they were entered into the regression.

### 3.6.1. Block 1

This included COVID-19 variables found to significantly impact differences in trust levels, including parents who did and did not work with their professionals during the pandemic, parents who indicated that communication impacted their communication positively, and not positively, and parents who were able to assess the impact of COVID-19 versus those who were not able to assess its impact. This block of variables was found to be statistically significant at predicting the parent trust of professionals ($R^2$ = 0.081, F(3,153) = 4.523, and *p* = 0.005; adjusted $R^2$ = 0.063). An examination of the coefficient statistics showed that a parent's ability to assess COVID-19's impact was a significant predictor of trust (β = −0.20, t = 2.46, and *p* = 0.006). The remaining variables, including parents who reported that COVID-19 positively impacted communication with their professional (β = 0.09, t = 1.09, and *p* = 0.280), and parents who worked with their professional during the pandemic (β = −0.09, t = 1.08, and *p* = 0.278), were not significant predictors of parent trust.

**Table 4.** Hierarchical multiple regression predicting trust from pandemic variables, specific learning disability, independent supports, legal measures, in-person and phone communication, and communication skills of special education professionals.

| Block | Variable | B | β | Std. Error | t |
|---|---|---|---|---|---|
| Block 1: Significant COVID-19 variables $R^2 = 0.081$ * | Constant | 60.14 ** | - | 3.64 | 16.53 |
| | Parents who worked with professional during COVID | −2.57 | −0.09 | 2.37 | −1.08 |
| | Parents who reported COVID had positive impact | 3.03 | 0.09 | 2.78 | 1.09 |
| | Parents who were unable to assess COVID's impact | −7.54 * | −0.20 | 3.06 | −2.46 |
| Block 2: Addition of parent groups expressing significantly lower or higher levels of trust $\Delta R^2 = 0.127$ ** | Constant | 57.89 ** | - | 3.53 | 16.39 |
| | Parents who worked with professional during COVID-19 | −4.14 | −0.15 | 2.25 | −1.84 |
| | Parents who reported COVID-19 had positive impact | 2.76 | 0.082 | 2.61 | 1.06 |
| | Parents who were unable to assess COVID-19's impact | 2.83 * | −0.22 * | 2.89 | −2.77 |
| | Parents of children with SLD | 2.164 | −0.11 | 2.16 | −1.45 |
| | Parents who used independent supports | 6.014 * | 0.213 * | 2.16 | 2.77 |
| | Parents who used legal measures | 5.950 * | 0.214 * | 2.15 | 2.77 |
| Block 3: Addition professional communication method $\Delta R^2 = 0.049$ * | Constant | 53.63 ** | - | 3.74 | 14.33 |
| | Parents who worked with professional during COVID-19 | −3.97 | −0.14 | 2.20 | −1.81 |
| | Parents who reported COVID-19 had positive impact | 2.27 | 0.07 | 2.58 | 0.88 |
| | Parents who were unable to assess COVID-19's impact | −8.07 * | −0.22 * | 2.82 | −2.85 |
| | Parents of children with SLD | −1.95 | −0.07 | 2.15 | −0.91 |
| | Parents who used independent supports | 5.67 * | 0.20 * | 2.11 | 2.68 |
| | Parents who used legal measures | 6.08 * | 0.22 * | 2.11 | 2.88 |
| | Professionals who communicated in-person | 4.73 * | 0.17 * | 2.10 | 2.26 |
| | Professionals who communicated over the phone | 3.87 | 0.14 | 2.02 | 1.91 |
| Block 4: Addition of professional's communication skills $\Delta R^2 = 0.351$ ** | Constant | 15.48 | - | 4.30 | 3.60 |
| | Parents who worked with professional during COVID-19 | −2.02 | −0.07 | 1.61 | −1.30 |
| | Parents who reported COVID-19 had positive impact | −1.38 | −0.04 | 1.91 | −0.72 |
| | Parents who were unable to assess COVID-19's impact | −3.73 | −0.10 | 2.09 | −1.78 |
| | Parents of children with SLD | −0.86 | −0.03 | 1.57 | −0.547 |
| | Parents who used independent supports | 2.67 | 0.09 | 1.56 | 1.71 |
| | Parents who used legal measures | 1.19 | 0.04 | 1.60 | 0.75 |
| | Professionals who communicated in-person | 4.13 * | 0.14 * | 1.53 | 2.70 |
| | Professionals who communicated over the phone | −0.30 | −0.01 | 1.52 | −0.19 |
| | Professional's communication skills | 0.49 ** | 0.68 ** | 0.04 | 11.48 |

N = 158. * $p < 0.05$, ** $p < 0.001$.

### 3.6.2. Block 2

This added the demographic characteristics of parents found to impact differences in trust and the significant COVID-19 variables that were entered in Block 1. These included

parents who did and did not identify themselves as having a child with an SLD, parents who did and did not indicate that they used supports independent of school, and parents who did and did not pursue legal measures to obtain services for their children. The addition of these variables led to a statistically significant increase in the prediction of parent trust ($R^2 = 0.208$, $F(3,150) = 6.579$, and $p < 0.001$; adjusted $R^2 = 0.177$; $\Delta R^2 = 0.127$). The coefficient statistics indicated that a parent's ability to assess COVID-19's impact continued to be a significant predictor of trust ($\beta = -0.22$, $t = 2.77$, and $p = 0.011$). Parents who obtained support independent of the school ($\beta = 0.213$, $t = 2.78$, and $p = 0.006$), and parents who pursued legal measures ($\beta = 0.214$, $t = 2.77$, and $p = 0.006$) also proved to be significant predictors of trust. Parents of students identified with an SLD did not significantly predict the trust of parents in this model ($\beta = 0.11$, $t = 1.45$, and $p = 0.149$).

### 3.6.3. Block 3

This included the significant COVID-19 variables from Block 1, the significant parent demographic characteristics from Block 2, and added communication methods that when present, or not present, indicated differences in trust. The communication methods included in-person and phone communication. The addition of the communication variables led to a statistically significant increase in the prediction of trust ($R^2 = 0.257$, $F(2,148) = 6.414$, and $p = 0.001$; adjusted $R^2 = 0.063$; $\Delta R^2 = 0.049$). The addition of in-person communication with professionals predicted the trust of parents ($\beta = 0.17$, $t = 2.26$, and $p = 0.026$). Phone communication with professionals did not predict the trust of parents ($\beta = 0.14$, $t = 1.91$, and $p = 0.058$). A parent's ability to assess the impact of the pandemic ($\beta = 0.22$, $t = 2.82$, and $p = 0.005$), parents who used supports independent of the school ($\beta = 0.20$, $t = 2.68$, and $p = 0.008$), and parents who used legal methods to obtain services for their children ($\beta = 0.22$, $t = 2.88$, and $p = 0.005$) also continued to be significant predictors of parent trust in this model.

### 3.6.4. Block 4

This added the significant COVID-19 variables from Block 1, the significant demographic characteristics from Block 2, the significant communication variables from Block 3, and added the variable of interest and professional communication skills, as assessed by the CSRS. The addition of this variable contributed to a statistically significant increase in the prediction of parent trust ($R^2 = 0.608$, $F(1,147) = 25.370$, and $p = 0.001$; adjusted $R^2 = 0.584$; $\Delta R^2 = 0.351$). An examination of the coefficient statistics indicated that the only significant predictors of trust were in-person communication with the professional ($\beta = 0.14$, $t = 2.70$, and $p = 0.008$), and the communication skills of the professional ($\beta = 0.68$, $t = 11.48$, and $p < 0.001$). All other predictors, including those that were significant in previous models were not significant predictors of trust in this model.

## 4. Discussion

Research on school communities strongly suggests that trust, regardless of its operational definition, is a protective factor that when present, signals positive outcomes for students, families, and schools. As has been stated previously, when higher levels of parent trust are found within school communities, schools are more likely to solve problems [28], involve parents [26,30] and demonstrate higher levels of academic achievement [21]. Alternatively, trust, or the lack thereof, may become a casualty when special education professionals and parents struggle to communicate effectively and manage conflict [11,42,43]. Special education professionals and those who are responsible for their professional development, such as administrators, mentors, and preservice instructors, therefore, should have an awareness of the factors that can predict this important outcome.

With these facts in mind, we undertook an investigation of the relationship between special education professionals' communication skills and the trust of parents of students with disabilities. We examined this relationship not only by evaluating parents' impressions of their professionals' communication skills and their trust of professionals, but also

attempted to identify other factors that could potentially impact the trust between parents and special education professionals. To this end, we created an electronic survey and administered it to parents living in the United States. The parents completed the survey during the latter half of the COVID-19 pandemic. One-hundred and sixty-five parent responses were obtained, and several factors that either related to or could differentiate the trust levels of parents emerged from our analyses. Before identifying these factors, it is worth revisiting the factors that had no statistical bearing on trust.

The parent's role, marriage status, employment status, socioeconomic status, race, state of residence, prior educational work experiences, the type of school that children attended, and the position of the professional were group factors that could not significantly differentiate the levels of parent trust. Significant differences in trust were not found based on whether the professional used video conferencing, email communication, and texting, or not. The parent age, parent education level, the number of children living in the home, and the number of children in the home receiving special education services did not relate to the trust of parents either. Surprisingly, the child's age did not have a significant relationship to parent trust in our study. This result appears to contradict the results obtained by Adams and Christenson [26] who found that parent trust declined significantly as children moved from elementary to secondary grades. That study had a much larger sample size (n = 1,234), was conducted within a suburban setting, was not conducted during a pandemic, and was not restricted to parents of students of disabilities, but rather to parents in general. Additional research may be needed to determine whether any of these factors, in isolation, or together can predict the trust levels of parents.

Based on parent preferences as reported in the literature, it was expected that the frequency that professionals communicated with parents would relate to the parent trust of the professional [11,47–55]. The findings of this study did not support this assertion. The frequency that parents communicated with their child's professional was not related to the trust of the professional. These results may make intuitive sense given the diversity of messages that professionals send, and that parents receive. If the professional's messages are frequent, but are received negatively or are viewed as incompetent, one could hardly expect the professional to secure parent trust. At the same time, it is premature to state with absolute certainty that communication frequency has no bearing on trust. It is possible that the communication frequency and trust relationship may weigh more heavily on the expectations of the parent. When parents ask for communication that is frequent and consistent [33], they may mean that they would prefer their professional(s) meet their personal expectations for communication. This assertion, however, is highly tentative. The amount of appropriate communication may depend on the needs of the child, and the needs and expectations of the parent, but additional research is necessary to clarify this discrepancy in the literature.

There were factors that did have a bearing on trust in our preliminary analyses. We entered these factors into regression analyses to determine which, if any, could help predict the trust of parents. An explanation of these factors follows.

The COVID-19 pandemic was found to be a significant determinant of the trust of parents. Our preliminary findings indicated that parents who worked with their child's professionals during the pandemic expressed higher levels of trust in their professionals than parents who did not work with their professionals during the pandemic. These parents gave significantly higher ratings of their professionals' communication than parents who did not work with their child's professional during the pandemic. Parents who reported that COVID-19 had a positive impact on their communication with their professionals reported higher levels of trust than those who did not report that COVID-19 had a positive impact. The results showed that parents who were not able to assess the impact of the pandemic expressed significantly lower levels of trust and gave lower ratings of their professionals' communication skills than those parents who were able to assess the pandemic's impact.

The full impact of the COVID-19 pandemic on parent–professional relationships is unknown, but it is perhaps unsurprising that its' influence was observed in the preliminary

results of this study. Some research has suggested that parents felt that the instruction provided to their children who received IEP services was "ineffective". Briesch et al. [52] evaluated the daily experiences of U.S. students from the perspective of caregivers during the pandemic. Compared to pre-COVID levels, the caregivers of children who received special education services reported decreases in satisfaction with the services their children received. The parents in that study cited "effective communication strategies" as both a factor that supported the learning experiences of their children, and as a factor that resulted in stress. Several caregivers in that study reported that clearer and more consistent forms of communication could have supported the family's experience. Given some of the negative experiences reported by parents, the pandemic may have reduced the opportunities for professionals to demonstrate effective communication skills, which may have reduced the opportunities for professionals to establish trusting relationships with parents.

There were certain group characteristics of parents that, when present, signaled lower levels of parent trust. This study showed that parents whose children were identified with an SLD, parents who used independent supports, and parents who used legal means to obtain services for their children were significantly less trusting of their professionals than those parents whose children were not identified with an SLD or who did not pursue legal means or use independent supports to obtain services for their children.

Although a clear explanation has not emerged from the literature, studies show that students identified with an SLD are represented in due process litigation at higher rates than other categories [53–55]. It is worth noting that the defining characteristics of SLDs have been debated since they were formally recognized under the Education for All Handicapped Act (EAHCA) in 1975 [56,57]. Given that parent–professional conflict can be the result of differing perspectives about the definitions of a disability [11], it is possible that ambiguity surrounding the definition of SLDs contributes to conflict among teams of parents and professionals. This explanation is, however, tenuous given the lack of available research. Research about how parents and professionals may differ in their conceptualizations of an SLD may help to identify potential areas of disagreement.

It is perhaps not surprising that those parents who pursued due process litigation or complaints would express lower levels of trust in their professionals, especially given the research that has found such avenues to obtain services for children to be adversarial and ineffective at resolving conflict [2,3,6]. Qualitative research has indicated that the legal mechanisms for resolving conflict, including mediation [8], and meeting facilitation [9], have both helped to repair parent–professional relationships. These interventions, however, require supports from third parties outside the relationship [2]. Research that examines the efficacy of interventions that target persons directly involved in parent–professional relationships may lead to solutions to conflicts that are more proactive and efficient.

Unlike other modes of communication, parents reported higher levels of trust when professionals contacted them over the phone or spoke to them in-person. Media richness theory (MRT) may help to explain these findings. This theory, as initially proposed by Daft and Lengel [58], arranges media on a hierarchy of rich to lean forms of communication. MRT suggests that rich forms of communication are most effective when individuals transfer messages that are uncertain or equivocal. Rich media provide immediate and personalized feedback, use several cues or channels, and allow for exchanges that use natural language. Rich media reduce the potential for conflicting messages, or a lack of information. In contrast, lean forms of communication are limited in their capacity to provide instant feedback, cannot transmit multiple cues, such as body language and tone of voice, and restrict the use of natural language and personal focus. MRT suggests that lean modes of communication are more appropriate and efficient when both parties have a mutual understanding of the message. MRT has received attention in the literature as technologies become increasingly prevalent in education, and parent–teacher interactions [59–61]. Thompson's [61] research shows that parents generally prefer email communication for its convenience. The parents and teachers in that study reported that email was a sufficient method for sharing information that requires little interpretation, such as schedules, and

grades [61]. When sharing information concerning a student's behavior, teachers expressed a greater hesitancy to share the information in an email, preferring to use another form of oral communication [61].

In this study, the richest form of communication was in-person communication because parents had an opportunity to give and receive feedback. Using this form of communication, the parents and professionals may have had the opportunity to ask questions and evaluate the nonverbal cues of one another. Video conferencing, and phone communication followed in-person communication in terms of richness, and our results showed that of the two, only phone communication signaled trust differences. Briesch et al. [52] found that parents did not have difficulty in accessing technology used for video conferencing during the pandemic, but its widespread use is recent compared to more traditional modes of communication. As the preferences for communication technology change, it may be useful to determine the extent to which its use can facilitate trust.

The lean forms of communication in our study included email and text messaging. Given the importance of sharing accurate and clear information in a parent–professional relationship [62,63], and the potential for communication in these relationships to be conflicting and unclear [64,65], it is reasonable to assume that parents have a greater opportunity to experience trust when professionals use rich communication media. Other forms of communication, while convenient and preferred, may not be as helpful in facilitating trust. Given the differences found between video conferencing and phone communication, MRT's application to the current study results may be tenuous, and as preferences change, future research may be necessary to determine whether professionals should select one communication medium over another to establish trust.

As expected, we found a significant, positive, and moderately strong relationship between a professional's communication skills and parent trust. The parent ratings of a professional's communication skills indicated that parents perceived it as a one-dimensional construct, as opposed to a multi-dimensional construct that has emerged in other groups of individuals [45,46]. The professionals who parents rated as highly attentive were also likely to receive high ratings regarding how well-composed, coordinated or expressive they were during a conversation. Ipso facto, parents who rated their professionals as having low coordination skills, also rated them as having low levels of attentiveness, composure, and expressiveness. Thus, communication skills as a one-dimensional construct served as a variable interest in our final regression model.

The significant indicators of trust were entered into a four-model hierarchical multiple regression. Groups of parents who did and did not indicate that they worked with their professional during the pandemic, who did and did not report that COVID-19 had a positive impact on their relationship, and who could and could not assess the impact of COVID-19 were entered into the first model. The groups of parents who did and did not identify as having a child with an SLD, who did and did not use independent supports, and those parents who did and did not use legal remedies to obtain services for their children were entered into the second. The groups of parents who reported that their professional spoke to them in-person or not, and those parents who reported that professionals contacted them over the phone were entered into the third. The ratings of a professional's communication skills were entered in the fourth model.

In the final model, the only significant predictors of trust were the professional's appropriate and effective use of communication skills, and the professional's use of in-person forms of communication. All other variables, including those that were significant in other models were not significant predictors of trust in the final model. In other words, a parent's judgement of competence of specific behaviors during an in-person conversation carried substantial weight regarding the parent's perceptions of a professional's trustworthiness, and even other factors that signaled a reduction in trust, or conflict were present, such as the pursuit of legal measures or the solicitation of independent services.

*Limitations*

There are several factors that limit the validity of the results we obtained in our study that are important to mention. One of the main limitations of our study is that we collected our data during the latter half of the COVID-19 pandemic when children were returning to school and families were returning to work. It is very difficult to determine whether our results can be generalized to the present day where most have returned to work and school. Support for our results may improve as future studies are conducted when parents, students, and professionals are following their typical routines.

The convenient sampling of parents also severely restricts the generalizability of our results to parents whose children attend public schools, who work in education, and who live in the northeast region of the United States. The parents who participated in our study presumably had access to technology to complete the online survey, which in turn left out parents who had reduced access to online technology. Most of the parents who entered the study identified themselves as mothers. Fewer numbers of parents identified themselves as fathers, grandmothers, or guardians. This trend of mothers' overrepresentation in the research involving special education research is consistent with previous research [13,39,64,65], and limits the generalizability of the results reported here to the mothers of students with disabilities. In addition, most parents indicated they were white, married, employed, and not living in distressed economic conditions. The average parent who completed the survey had at least the equivalent of a four-year college degree. At least half of the sample included parents who had prior work experiences in education, and while parents with diverse races and economic backgrounds, and non-educators were included, researchers in the future should attempt to obtain a more diverse sample of parents to determine the importance that communication competence has for other populations. Specifically, future studies could include large numbers of parents and professionals of color and examine how a racial–ethnic match/mismatch between families and special education professionals impact trust and communication. The research indicates that the race of children can influence the quality of teacher–child relationships [66] and it is theorized that a teacher–child ethnic match has explicit benefits for children, especially for children of color [67]. These benefits of racial match presumably extend to other education professionals; however, empirical research is needed to confirm this.

We used a free and reduced lunch price (FRLP) as a variable to indicate the socioeconomic status, instead of asking parents directly about their household income. Prior studies that have examined parent trust in educational contexts have used FRLP as a proxy for SES [20,23,27], but FRLP use has received criticism due to concerns that it biases inferences made about income and poverty and does not provide a valid reflection of wealth or access to financial resources [68,69]. In a study of schools in California and Oregon, Domina et al. [70] found that the use of FRLP categories was unsuitable for representing the variation in household income, but they also found that FRLP predicted academic achievement more robustly than IRS-reported income data. Given the inaccuracy of FRLP as an income variable, future studies should consider collecting information on household income as a continuous variable. Collecting information on FRLP, however, may be suitable if academic achievement is a variable of interest.

The ecological validity of this study's results is also limited given that the CSRS does not include an exhaustive list of behaviors that can be central to appropriate and effective communication between various interactants, or among various groups or cultures [45]. Spitzberg suggests, for example, that the CSRS does not include the "use of silence" as a behavior that is important in Asian cultures [46] p.16. The transparency of a professional's language, such as the professional's use of specific words or phrases, may be important to consider in parent–professional interactions, given research showing that a professional's overuse of educational or psychological jargon can inhibit communication [9,42,63]. The professional's ability to maintain and not deviate from important topics of conversations may also be critical to a parent's judgment of competence based on research suggesting that

parents prefer professionals who emphasize the child in conversations over conversations where paperwork is the primary focus [42–44].

Although we gathered information about parents' use of legal supports and independent services, we did not consider how parents' previous experiences with professionals impacted their trust toward their current professional. As the literature indicates, prior negative experiences can have a lasting impact on parent perceptions [8,9,12]. Given the potential for confounding the perceptions of previous professionals with current professionals, items regarding previous experiences were left out, but having direct knowledge of these prior experiences may have improved the predictive capacity of our model. Future research should consider evaluating how previous experiences with professionals influence current parent–professional relationships.

We used a hierarchical multiple regression design to predict the trust of parents. Because this type of design provides information about a relationship between a set of variables, our results should not imply causation. In other words, one cannot use our study to assume that a professional's communication skills influence the trust of parents. Nor could we determine how other factors in our study, such as COVID-19, influence the trust of parents. To address this limitation, studies in the future should consider the use of experimental procedures or procedures that involve control and experimental groups with random assignments to determine the extent to which interpersonal communication skills can influence the trust of parents.

Our regression analyses uncovered seven parent responses that were outliers. These sets of responses were removed, which may limit the generalizability of our findings. These individuals were rated on both the higher and lower ends of the trust distribution. When the independent variables found to be significant were examined, all seven parents indicated that they worked with their professionals during the pandemic. All indicated that they had used an independent evaluator to secure services for their child. The data did not reveal other common characteristics; therefore, it is difficult to determine if there was an underlying characteristic that would explain their responses. As a result, it is unclear what their significance was.

Our design also used disability categories that are used in some states but are not identified at the federal level under IDEA (2004). This limitation may have inadvertently created confusion among parents in their selection of their child's disability category. For example, orthopedic impairments, traumatic brain injuries and multiple disabilities were not selections offered to parents. Parents whose children were identified with these disability categories may have been forced to select an inexact category match (e.g., a neurological disability instead of a traumatic brain injury) or may have chosen not to participate in the study. It is difficult to determine the extent to which this limitation impacted our study. Future studies that conduct research within a particular U.S. state are advised to use state-designated categories. Studies that are conducted between states, such as ours, are advised to use disability categories that fall under IDEA.

## 5. Study Implications

For systems to deliver effective special education services, stakeholders must consider the costs associated with providing essential resources. Perhaps the most important resource in an effective system is the depth of skill that each professional can provide. Research suggests that when professionals lack the skills necessary to maintain trust in a parent–professional relationship, the parents of students with disabilities, schools, and professionals can suffer financially and emotionally [3,5,6]. Litigation expenses can strain budgets [3,5,11,71–73], and professionals may leave the profession early due to stress from protracted legal disputes [6]. The litigation that some districts face is unsustainable, especially given the shortages and high attrition rates that have been found among special education professionals [74–76]. The results obtained in the current study suggest that professionals who are ineffective and inappropriate in conversations with parents may experience greater difficulty in their efforts to obtain trust. Furthermore, special education

professionals who are judged to be poor communicators may be susceptible to the negative outcomes associated with reduced parent trust, especially when parents have engaged in due process complaints and litigation, have used services independent of the school, have children who have been identified with an SLD, and/or those parents who have had unfavorable experiences with schools due to COVID-19. On the other hand, professionals who can communicate appropriately and effectively in conversations with parents are more likely to experience the benefits of a trusting relationship. These results comport with studies suggesting professional communication is a factor implicated in the escalation or de-escalation of a conflict [11–13].

It has been suggested that professionals need preparation to work skillfully within a parent–professional relationship [77,78], yet research suggests that preservice professionals are entering the field ill-prepared to work collaboratively with parents [79]. Our research suggests that favorable parent judgments of a professional's attentiveness, composure, expressiveness, and coordination in conjunction with in-person forms of communication, predicts the trust of parents, even when factors that can reduce it are present. A combination of opportunities for practice, effective mentorship, and learning the components of partnerships may accelerate a professional's development of these skills. Hampshire et al. [80], for example, explored the experiences and perspectives of preservice special educators who received training in development of family professional partnerships (FPP). FPPs are based on seven fundamental principles that include: (a) communication, (b) professional competence, (c) respect, (d) commitment, (e) equality, (f) advocacy and (e) trust [41,77]. With the experiences of providing direct services to children and families, receiving supervision from skilled special educators and helpful feedback from parents, the preservice providers in Hampshire et al.'s study [80] overcame fears of appearing incompetent, confronted negative stereotypes of families, learned to listen more effectively, and acquired respect for the value parents brought to relationships. It is possible that professionals could strengthen their communication skills through similar experiences that involve direct contact with parents, supervision from special educators who are skilled at communication, and parent feedback, but additional research would be needed to make this assertion.

**Author Contributions:** Conceptualization, J.E.H.; methodology, J.E.H. and A.D.M.; software, J.E.H.; validation, J.E.H. and A.D.M.; formal analysis, J.E.H. and A.D.M.; investigation, J.E.H.; resources, J.E.H.; data curation, J.E.H.; writing—original draft preparation, J.E.H.; writing—review and editing, J.E.H. and A.D.M.; visualization, J.E.H. and A.D.M.; supervision, J.E.H. and A.D.M.; project administration, J.E.H. and A.D.M.; funding acquisition, J.E.H. and A.D.M. All authors have read and agreed to the published version of the manuscript.

**Funding:** This research received no external funding.

**Institutional Review Board Statement:** The study was conducted in accordance with the Declaration of Helsinki, and approved by the Institutional Review Board of Rivier University (protocol code 092721, date of approval: 10 November 2021).

**Informed Consent Statement:** Informed consent was obtained from all subjects involved in the study.

**Data Availability Statement:** Not applicable.

**Conflicts of Interest:** The authors declare no conflict of interest.

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
