# Peer review of "Predicting Parent Trust Based on Professionals’ Communication Skills"

_education, doi:10.3390/educsci13040350_

Round 1

Reviewer 1 Report

In this article, the authors present the results of a study on the relationship between parents' evaluation of professionals' communication skills and parents' trust in professionals. 165 parents responded to an online survey during the COVID-19 pandemic

The title of the manuscript is quite concise and manages to focus attention on an object of study that, once read in more depth, seems to be the main one or, simply, the one that is addressed to a greater extent throughout the article. .

The abstract of the article clearly justifies the need or importance of carrying out a study of this nature, and the main objectives, results and most significant conclusions of the study, as a result of reading the abstract of the article, are also very clear and structured. The ideas appear perfectly organized, structured and contextualized, which provides the reader with a broad overview of the most significant arguments and information that are handled in this section of the article.

The main scientific background of the problem under study is continuously mentioned.

The paper is interesting and has the potential to contribute to the existing body of emerging literature. However, there are some areas of the document that should be strengthened.

First, the document would benefit from providing a clear definition of the research problem. Regarding the methodology, the study design should be clarified. More information on the data collection procedure and sample would also be appreciated. The Discussion section is vague and generic. This is where the essence of the study and the recommendations must be presented in a coherent way. I propose that the biases that could be produced by the sample be analyzed a little more in depth, taking into account that said sample is biased (mainly mothers, white race,...)

I wish the authors the best of luck in advancing their research and in crafting their contributions to the literature.

Author Response

To whom it may concern:

Thank you for your review and sound commentary on our manuscript, “Predicting Parent Trust based on Professionals’ Communication Skills.” We genuinely appreciate the feedback!

Our response to your comments is as follows.

Based on your recommendations we reworded our research problem in the form of a research question and posed a research and a null hypotheses in our introduction. We added details about our data collection procedures in the materials and methods section. We also added a data analysis section that describes how we went about analyzing our data. We added tables to help readers readily identify characteristics of our population.

We expanded the discussion section in several areas. To contextualize the importance our study, and to make it less generic, we added an introductory paragraph that describes the importance of trust in parent-professional relationships. You will also see that we expanded our limitations section due to the generally homogenous sample we obtained. We also made recommendations for future research.

We highlighted where we made changes.

Again, we thank you for your feedback on our study!

Reviewer 2 Report

This study aimed to investigate the relationship between parent evaluation of professionals’ communication skills and parent trust of professionals as well as the influence of COVID-19. Though the study was conducted during the pandemic, I have doubts whether findings of this study could proof the impact of COVID-19 as there was no comparison group, nor it was a longitudinal study.

In page 6, authors mentioned that “We used dummy variables to run independent t-tests to determine whether parents of the positive impact group differed in their levels of trust from parents who did not report a positive impact. Findings indicated that parents who reported that COVID-19 had a positive impact on their communication trusted their child’s professional to a significantly greater extent than parents who did not report that COVID-19 had a positive impact”. More explanation is needed to describe what the “dummy variables” are, and how this justify the methodology. Moreover, there were a number of “COVID-19 variables” but they were not seem to be homogenous, i.e., “Parents who worked with professional during COVID”, “Parents who reported COVID had Positive impact”, “Parents who were unable to assess COVID’s impact”. How these variables were defined and being measure were unclear.

 Authors define trust as “…confidence that another person will act in a way to benefit or sustain the relationship, or the implicit or explicit goals of the relationship, to achieve positive outcomes for students.” (page 6). Please add more discussion at the beginning of the paper to explain the concept of “trust” in professional service, as comparing with “confidence”. These two terms are not interchangeable. Readers might also want to know more about the research gap in professional trust, especially in service for people with disability. More literature should be cited to support your argument.

 Authors might have reported too many findings, e.g., the results of t-tests and ANOVAs. Consider choosing only those which are essential to report. It could make the paper less clumsy.

Author Response

To whom it may concern:

Thank you for your review and sound commentary on our manuscript, “Predicting Parent Trust based on Professionals’ Communication Skills.” We genuinely appreciate the feedback!

Based on your comments, we included research in the introduction that clarified the concept of trust, how it has been described in the literature, and a rationale regarding why more research has been needed on parents of students with disabilities and their child’s special education professionals.

We added clarification about how we collected data from parents on their perceptions of COVID-19 in the form of two questions in the data collection section of our manuscript. We also indicated how we coded the variables so that we could run t-tests in regard to COVID-19 variables in our results section. We expanded our limitations section to describe how our design could not prove the influence of COVID-19 on parent trust.

Based on your recommendation, we eliminated some of the ANOVA/t-test analyses that we ran, especially if they were not significant.

We highlighted where we made changes.

Again, we thank you for your feedback on our study!

Author Response

To whom it may concern:

Thank you for your review and sound commentary on our manuscript, “Predicting Parent Trust based on Professionals’ Communication Skills.” We genuinely appreciate the feedback!

Our response to your comments is as follows.

Introduction.

We altered the purpose statement of our study and mentioned COVID-19 as a variable that we set out to understand as a factor that may have an impact on parent trust. We added null and research statements to our manuscript to clarify the purpose of our study.

Materials and Methods.

We added information about the recruitment process and created additional tables that described our participants and their children. We organized the thirty states represented in our study by geographic region and provided the top 5 states represented in our study. We also added information about parents work experiences in education and created a demographic table.

We chose to use Free and Reduced Lunch Price (FRLP) to collect data on socioeconomic status because previous studies on trust have used it as a proxy for SES, and we also hoped to minimize the possibility that we would have missing data in our results. We added a paragraph discussing the debate and limitations surrounding the use of FRLP in our limitations section.

Child Characteristics.

We added SD to child’s age, age at which children were initially identified with a disability, and reworded phrasing that described their age in terms of years and months.

We describe how the disability categories were selected (i.e., based on the state level categories in which the authors work). We acknowledge that this may have created confusion among parents who responded to our survey, and we discussed this in the limitations section of our manuscript.

Data Collection.

We changed phrasing around the CSRS based on your recommendation, and provided information about the FSRS and CSRS in terms of their reliable use in previous studies. We also mentioned how our use of the CSRS has been used in a similar manner in previous research, and described how we adjusted it for our purposes. We also added a Data Analysis section to describe how we would determine which variables we would use in our regression analysis.

Results/Discussion.

We changed the headings so that the focus of each variable would be its pertinence to our dependent variable trust. Our tables and our discussion delve further into the limitations of our study’s generalizability than we had described previously. We provided additional detail on the outliers in our study, how we defined them, and added them as a discussion point in our limitations section. We reviewed the citations carefully and made adjustments where found that they were incomplete.

We highlighted where we made changes.

Again, we thank you for your feedback on our study!
